# Evaluation of a Medical Grade Thermoplastic Polyurethane for the Manufacture of an Implantable Medical Device: The Impact of FDM 3D-Printing and Gamma Sterilization

**DOI:** 10.3390/pharmaceutics15020456

**Published:** 2023-01-30

**Authors:** Marie-Stella M’Bengue, Thomas Mesnard, Feng Chai, Mickaël Maton, Valérie Gaucher, Nicolas Tabary, Maria-José García-Fernandez, Jonathan Sobocinski, Bernard Martel, Nicolas Blanchemain

**Affiliations:** 1Univ. Lille, INSERM, CHU Lille, U1008—Advanced Drug Delivery Systems and Biomaterials, F-59000 Lille, France; 2Univ. Lille, CNRS, INRAE, Centrale Lille, UMR 8207—UMET—Unité Matériaux et Transformations, F-59000 Lille, France; 3Institut Coeur Poumon, Regional Hospital Center University of Lille (CHRU Lille), 2 Avenue Oscar Lambret, F-59000 Lille, France

**Keywords:** medical device, 3D printing, polyurethane, sterilization, biocompatibility

## Abstract

Three-dimensional printing (3DP) of thermoplastic polyurethane (TPU) is gaining interest in the medical industry thanks to the combination of tunable properties that TPU exhibits and the possibilities that 3DP processes offer concerning precision, time, and cost of fabrication. We investigated the implementation of a medical grade TPU by fused deposition modelling (FDM) for the manufacturing of an implantable medical device from the raw pellets to the gamma (γ) sterilized 3DP constructs. To the authors’ knowledge, there is no such guide/study implicating TPU, FDM 3D-printing and gamma sterilization. Thermal properties analyzed by differential scanning calorimetry (DSC) and molecular weights measured by size exclusion chromatography (SEC) were used as monitoring indicators through the fabrication process. After gamma sterilization, surface chemistry was assessed by water contact angle (WCA) measurement and infrared spectroscopy (ATR-FTIR). Mechanical properties were investigated by tensile testing. Biocompatibility was assessed by means of cytotoxicity (ISO 10993-5) and hemocompatibility assays (ISO 10993-4). Results showed that TPU underwent degradation through the fabrication process as both the number-averaged (Mn) and weight-averaged (Mw) molecular weights decreased (7% Mn loss, 30% Mw loss, *p* < 0.05). After gamma sterilization, Mw increased by 8% (*p* < 0.05) indicating that crosslinking may have occurred. However, tensile properties were not impacted by irradiation. Cytotoxicity (ISO 10993-5) and hemocompatibility (ISO 10993-4) assessments after sterilization showed vitality of cells (132% ± 3%, *p* < 0.05) and no red blood cell lysis. We concluded that gamma sterilization does not highly impact TPU regarding our application. Our study demonstrates the processability of TPU by FDM followed by gamma sterilization and can be used as a guide for the preliminary evaluation of a polymeric raw material in the manufacturing of a blood contacting implantable medical device.

## 1. Introduction

3D-printing (3DP) designates processes aiming to build real three-dimensional objects from a computer–aided design. Long used in the automotive and aeronautic industries as a rapid prototyping technique, 3DP is now extended to other fields, including the biomedical and pharmaceutic domains. Various thermoplastic polymers were initially implemented by 3DP for the manufacturing of biomaterials such as polylactic acid (PLA), polycaprolactone (PCL), poly(lactic-co-glycolic) acid (PLGA) or poly(methyl methacrylate) (PMMA) [1,2,3,4,5]. Multiple medical specialties were impacted, from tissue engineering to orthopedic surgery, dentistry, cardiovascular and maxillofacial surgeries [4,6,7,8,9]. With recent technology advances, more polymers with unique properties are being implemented by 3DP processes for niche applications. Indeed, families of polymers such as polyaryletherketones (PAEK) [9,10,11] and thermoplastic elastomers (TPE) [12,13] are gaining increasing interest for their particular mechanical properties. Moreover, specialty polymers composites (bilayer encapsulated PCL-TPU) with shape-memory properties are being designed and implemented in 4D-printing, which introduces the concept of shape-shifting 3DP constructs under external stimuli (light, temperature or pH environment) [14].

Thermoplastic polyurethanes (TPU) are macromolecules based on aliphatic polyether segments that present high mobility (called soft segments, SS) separated by rigid aromatic groups (called hard segments, HS) that can organize by forming hard domains and crystallize. Indeed, below their melting temperature, TPU are biphasic materials made of (more or less semi-) crystalline clusters where HS are gathered by π-π stacking interactions forming hard domains, spread in the amorphous phase where chains of SS are entangled, forming soft domains [15]. Such specific organization provides the elastomeric properties of TPU. Depending on their nature—the SS to HS ratio and the degree of separation between these two phases—the properties of TPU can be tailored. From a biological standpoint, TPU is an excellent candidate as it exhibits biocompatibility and hemocompatibility [16,17]. It is used as a raw material for extracorporeal and implantable blood-contacting medical devices such as blood pockets, catheters, vascular grafts or arteriovenous shunts [18,19]. Therefore, the implementation of TPU by 3DP processes is a topic of growing interest. The wide range of properties offered by TPU coupled with rapid manufacturing, low-cost and a high-resolution technique, make an interesting combination for the medical industry. Some studies reported operating conditions of TPU by fused deposition modelling (FDM) [20,21,22]. It is a widely known 3DP process and its rationale consists of building 3D constructs by successive layering of melted polymer.

In the context of the fabrication of implantable medical devices, a polymer-based implant is to be sterilized as part of the manufacturing process. Common sterilization methods, such as gas sterilization with ethylene oxide or steam, irradiation sterilization using gamma rays or electron beam, are used in order to eliminate living microorganisms and avoid infections. The effects of sterilization using ionizing radiation on TPU have been thoroughly reported in the literature [22,23,24,25,26,27]. Chain scissions, or the rupture of bonds, and crosslinking, or the formation of bonds, are the two main phenomena identified, along with branching, in numerical studies [22,23,24,26,27,28]. These occurrences are material-sensitive and often times can occur simultaneously, leading to changes in chemical, thermal and mechanical properties of the polymer. Depending on the sterilization process, the intensity of the dose received and the nature of the material, these phenomena compete with each other, and as a result, TPU can undergo embrittlement, toughening, hardening or softening and discoloration [22,23,24,26,27,28]. The medical industry also requires a strict monitoring of the fabrication process for reliability and reproducibility concerns. Indeed, property changes, when they occur, must be identified and limited to guarantee safety and quality of the devices to be implanted [5]. Clinical performances are obviously dependent on property changes; therefore, it is of high importance to assess the effects of the fabrication process and their extent.

The purpose of this study was to evaluate the effect on TPU of a melt extrusion 3DP technique and a common sterilization method as a fabrication route for an implantable medical device. This study was completed with a preliminary assessment of the biological properties of the final sterilized samples. We investigated the effect of FDM on a medical grade TPU from raw pellets to gamma (γ) sterilized 3D parts. Molecular weights measured by size exclusion chromatography (SEC) and thermal properties assessed by differential scanning calorimetry (DSC) were used as monitoring indicators at each step of the process. Surface properties of sterilized 3D parts were investigated by means of water contact angle measurement (WCA) and infrared spectroscopy (ATR-FTIR) and mechanical properties by tensile testing. Furthermore, biological evaluation was conducted through cytotoxicity and hemocompatibility assays according to ISO10993-5 and ISO10993-4 standards, respectively.

To the authors’ knowledge, there is no such study that (i) provides data on the effect of FDM 3D printing followed by gamma sterilization on the physico-chemical properties of a medical grade TPU; and (ii) assesses the cytotoxicity and hemocompatibility of FDM processed and sterilized TPU constructs.

## 2. Materials and Methods

### 2.1. Materials

Medical grade TPU, Elastollan^®^ supplied by BASF (Lemförde, Germany), was received in raw pellets form. It is a polyether based TPU with polytetramethylene oxide (PTMO) as SS, 4,4-diphenylmethane diisocyanate (4,4-MDI) as HS and 1,4-butane diol (1,4-BDO) as a chain extender. Tetrahydrofuran (THF) of HPLC grade (≥99.8%) was supplied by Honeywell Riedel-de HaënTM (Seelze, Germany). Sodium Carbonate (Na_2_CO_3_) and salt tablets for Phosphate Buffer Solution (PBS) preparation were supplied by Sigma Aldrich (Steinheim, Germany). Ethanol of HPLC grade (≥99.9%) was supplied by Fisher Chemical (Geel, Belgium).

### 2.2. Samples Preparation

#### 2.2.1. 3D-Printing (FDM)

The manufacturing process of the TPU parts is summarized in Figure 1. In order to carry out FDM, the raw TPU pellets were extruded (Composer 350, 3devo, The Netherlands) into a 1.75 mm diameter filament. The optimum for temperature profile was determined in order to obtain a clean filament of regular diameter. Extrusion temperatures from the feeding to the extrusion nozzle were 155 °C, 175 °C, 181 °C and 185 °C according to the recommendations of the fabricant (BASF, Lemförde, Germany). The rotation speed of the single extrusion screw was 2.8 RPM. Tubular samples (height: 25 mm; diameter: 20 mm; wall thickness: 0.8 mm) were designed using an open-source software (On Shape, Boston, MA, USA), then processed for slicing (Simplify3D, Cincinnati, OH, USA) before being 3D printed by FDM (Stream 20 Dual MK2, Volumic, France) from the above-mentioned filaments following parameters in Table 1.

#### 2.2.2. Sterilization

FDM samples underwent gamma (γ) irradiation at a dose of 40 kGy according to the ISO 11137-2 to sterilized implantable medical device in vascular position. The minimum level dose validated using this process to sterilize medical devices is 25 kGy by exposition to an ionizing radiation generated by a cobalt 60 (^60^Co) source. This dose guarantees a probability of viability of a microorganism lesser or equal to 10^−6^.

### 2.3. Samples Characterization

#### 2.3.1. Water Contact Angle Measurement (WCA)

The contact angle measurements were carried out using a goniometer (Minitec DSA100, Krüss, Germany) equipped with an optical system allowing the capture of the drop and the analysis of the contact angle. Using a syringe, a drop of 2 μL of distilled water is deposited on the surface of a sample using the “Sawsen Sessile Drop” method. The measurements were taken on different areas of the sample in order to take into account its heterogeneity. Six measurements were carried out on each sample at room temperature. The angle formed by the baseline, i.e., the surface of the FDM sample and the tangent to the drop contour at the three-phase point (solid-liquid-gas) was our contact angle.

#### 2.3.2. Infrared Spectroscopy (ATR-FTIR)

The FDM samples were analyzed by Fourier transform infrared spectroscopy (FTIR) according to the attenuated total reflectance (ATR) method. The study was conducted using an IR spectroscope (SpectrumTwo, Perkin Elmer, France) and its associated software of the same name (Spectrum version 10.6.0). The study was carried out at room temperature and three spectra per samples studied were recorded in absorbance mode after 16 scans over a wave number range from 4000 cm^−1^ to 450 cm^−1^ with a resolution of 2 cm^−1^. The values were exported from the software in ASCII format in order to trace the absorbance spectra in Excel.

#### 2.3.3. Differential Scanning Calorimetry (DSC)

Differential scanning calorimetry (DSC Q100, TA Instruments, Guyancourt, France) was performed on the samples. About 5 mg of sample were sealed in a standard aluminum capsule in a chamber under inert atmosphere (N2, 0.8 bar) delivering a flow of 50 mL·min^−1^. An empty standard aluminum capsule was used as a reference. The thermograms of heat flow as a function of temperature were recorded over a temperature range between −65 °C and +260 °C on the basis of a heating-cooling-heating cycle with a ramp of 10 °C·min^−1^. Three thermograms were performed per sample. The glass transition temperatures Tg were taken as the midpoint temperatures and melting temperatures were taken as the maximum peak melting endotherm.

#### 2.3.4. Size Exclusion Chromatography (SEC)

The size exclusion chromatography was carried out using a WATERS E2695 chromatograph (Waters, Waters Corporation, Milford, MA, USA) equipped with HR-1, HR-3 and HR-4 columns (500–500,000 g·mol^−1^) and coupled with a differential refractometer (Optilab^®^-T-rEX, Wyatt Technology, Santa Barbara, CA, USA). The system is calibrated using a standard polystyrene solution. 15 mg of sample was dissolved in 3 mL of THF (n = 6). 1mL of toluene per liter of THF was added. Subsequently, these test solutions were filtered through a PTFE membrane with a pore diameter of 0.45 μm and then transferred to glass vials. Data processing was carried out using Astra 6 software (Astra 6, Wyatt Technology, USA).

#### 2.3.5. Tensile Tests

Dumbbells test pieces (12 mm × 4 mm) were cut in the parallel and perpendicular directions to the printing layers of samples. The test bench used was a universal traction machine (Instron 4466, Norwood, MA, USA) equipped with a climate chamber at 37 °C. The tensile tests were carried out with an initial strain rate of 3.5 × 10^−3^·s^−1^ until the rupture of the sample. The tests were carried out in triplicate and according to the ASTM D638 standard relating to the tensile properties of plastics.

### 2.4. Biocompatibility Assessment

#### 2.4.1. Cytotoxicity

The potential cytotoxicity of our biomaterial was evaluated by the extraction method (indirect contact) according to the ISO 10993-5 standard with a human pulmonary microvascular endothelial cell (HPMEC). FDM printed rectangular samples (with or without γ sterilization) with a surface area of 1.6 cm2 were soaked in 1.066 mL of complete culture medium (CCM) of HPMEC cells for 72-h extraction under agitation at 80 rpm and 37 °C on a Innova40 shaking incubator (New Brunswick Scientific, Illkirch, France). Each group of samples was triplicated. On the day before adding extraction medium, a 96-well tissue culture plate containing 100 µL of CCM and 4.0 × 10^3^ HPMEC cells per well was incubated at 37 °C for 24 h in a 5% CO_2_ atmosphere. After 24 h, the extraction medium was firstly filtere-steriled (0.22 µm Filtropur^®^, Sarstedt, Nümbrecht, Germany). Then, the culture medium was removed from the monolayer of cells and 100 µL per well of the filtered extraction medium or CCM (negative control, i.e., lack of cytotoxicity) were added.

The cell viability was assessed by the AlamarBlue^®^ test method after 24-h exposure to extraction medium. The Alamarblue^®^ solution, initially blue and oxidized, turns red and fluorescent when reduced by the enzymes associated with the respiratory activities in any eukaryotic cell. Briefly, the extraction medium was removed from the cells and 200 µL per well of a 10% AlamarBlue^®^ (ThermoFisher Scientific, Illkirch, France) in CCM were added and incubated at 37 °C, protecting from light, for 2 h. Then, 150 µL of the AlamarBlue^®^ solution from each well were correspondingly transferred into a black clear bottom 96-well plate for measuring the fluorescence at an excitation wavelength of 530 nm and an emission wavelength of 590 nm, on a microplate fluorometer (TwinkleTMLB 970, Berthold Technologies GmbH & Co, Wildbad, Germany). The results were then normalized relative to that of the negative controls (CCM) to express the relative cell viability of the test group.

#### 2.4.2. Hemocompatibility Analysis

For the test below, fresh human blood was collected by venipuncture with a 19 Gauge butterfly needle from a healthy adult volunteer (age 26) anticoagulated with 1.5 IU heparin/mL (Leo Pharmaceutical Products BV, Weesp, The Netherlands).

#### 2.4.3. Hemolysis

When in contact with a biomaterial, the interaction between chemicals or leachables and red blood cells can cause their destruction (lysis) and the release of intracellular hemoglobin, a principal constituent of red blood cells and responsible for their red color. Therefore, an in vitro hemolysis test was performed to assess the potential impact of material on red blood cells. FDM printed rectangular samples (with or without γ sterilization) with a surface area of 2.4 cm^2^ were pre-conditioned with 1 mL of PBS under agitation at 80 rpm for 30 min at 37 °C. Each group of samples was triplicated. The PBS without sample was used as a negative control, i.e., absence of hemolysis and 0.1% Na_2_CO_3_ without sample was used as a positive control, i.e., fully hemolysis. After conditioning, the sample membranes were then soaked in 1 mL per sample of human whole blood and were then incubated at 37 °C under agitation at 80 rpm for 1 h. Then, the blood was collected from each well and transferred to corresponding glass test tubes for 5-min centrifuging at 500× *g*. Hemolytic activity of samples was determined by measuring the absorbance of the collected supernatant from centrifuge tube at a wavelength of 541 nm (i.e., absorbance peak of free hemoglobin) using a UV spectrometer (UV-1800, Shimadzu, Marne La Vallee, France). Hemolysis rates were calculated following this formula:$$\%\;hemolysis=\frac{A_{sample}-A_{PBS}}{A_{Na_{2}CO_{3}}}\times 100\;\left(\phi \right)$$

#### 2.4.4. Blood Cells Adhesion

After contacting with the human whole blood in above test, FDM printed samples were transferred into a new 12-well tissue culture plate. For pre-treatment of SEM observation, each sample was soaked in 1 mL per well of a 2.5% glutaraldehyde solution for fixation for 30 min at 4 °C. The samples were then gradually dehydrated using ethanol solutions at different concentrations: 50%, 60%, 70%, 80%, 90% and 100%. Surfaces of each sample, metallized beforehand with a platinum coating (5 nm thick), were observed under a FlexSEM1000 scanning electron microscope (Hitachi, France) at an electron acceleration voltage of 5 kV, an emission current of 10 μA and a magnification of ×300.

### 2.5. Statistical Analysis

Results are presented as mean ± SD. The statistical analysis was carried out using Excel. To compare two groups, a two-way Student test (Student *t*-test) was used. To compare several groups to a control, a one-way analysis of variance (ANOVA one-way) is performed. The difference between two or more groups is considered significant when *p* ≤ 0.05.

## 3. Results

### 3.1. Impact of FDM Manufacturing Process and Sterilization on TPU

Molecular weights (Mw, Mn) and polydispersity index (Ip) from SEC (Figure 2) were used to investigate the impact of the manufacturing process of TPU followed by sterilization. Firstly, Mw dropped from 155,000 ± 14,000 g/mol to 108,000 ± 900 g/mol (cumulative decrease of 30%, *p* < 0.05) while Mn decreased from 68,000 ± 6000 g/mol to 58,000 ± 3800 g/mol (cumulative decrease of 15%, *p* < 0.05) from pellets to 3D printed form. This drop in molecular weight can be attributed to degradation occurring during extrusion of the raw TPU pellets into a filament and then during the 3D printing processing. After γ-sterilization, Mn remained stable at 59,000 ± 3400 g/mol while Mw increased up to 117,000 ± 2800 g/mol (8% increase, *p* = 0.0002). This growth could be the consequence of a crosslinking between macromolecular chains. Nevertheless, TPU remained soluble in THF after sterilization.

The DSC of the first heating and the subsequent cooling cycle of TPU of a pellet, a filament, a 3D printed, and a gamma sterilized sample, are shown in Figure 3. The first inflection point in the range of −50 °C to −40 °C, is associated with the glass transition temperature (Tg) of the soft domains of TPU. The second inflection point around 65 °C corresponds to the relaxation of the hard segments [28]. Collected data from DSC are shown in Table 2. Tg1 slightly increases after the extrusion step from −50 ± 2 °C to −45 ± 1 °C. Tg2, not apparent for the raw pellets, is detectable at 64 ± 3 °C after the extrusion step and then drops to 52 ± 2 °C (*p* = 0.03) after FDM processing. Moreover, Tm remains constant from 164 ± 1 °C to 163 ± 1 °C (*p* = 0.04) while ΔHm slightly decreases from 13.8 ± 0.5 J/g to 11.2 ± 0.5 J/g (*p* = 0.0004). After sterilization, Tg1 value remained unchanged contrary to Tg2 that rose noticeably to 61 ± 3 °C. Tm slightly increased from 163 ± 3 °C to 165 ± 1 °C while variation of ΔHm value was found non-significant (*p* = 0.68).

The impact of sterilization on the mechanical properties of the FDM samples was investigated through tensile testing (Figure 4). Collected data from tensile curves are summarized in Table 3. 3D printed samples exhibited elastomeric behavior with a Young’s modulus of 14.1 ± 1.2 MPa, elongation at break of 697 ± 17%, upon stretching in the parallel direction of the extruded filament and dropped down to 378 ± 15% in the perpendicular direction of the extruded filament. Indeed, interlayer adhesion interfaces constitute weak spots where failure is initiated and propagated. After γ sterilization, the tensile properties of TPU did not dramatically change as Young’s modulus, stress at break and strain at break slightly increased, respectively, by 4%, 11% and 4% in the parallel direction and by 1%, 8% and 17% in the perpendicular direction to the printing layers. Statistical analysis showed that those differences were not significant (*p* > 0.05).

### 3.2. Surface Properties of Sterilized 3DP Samples

After sterilization, the 3D printed samples originally transparent turned yellow. Water Contact Angles (WCA) measurement showed that the TPU sample exhibits a slight decrease of surface hydrophobicity as WCA decreased from 97.2 ± 1.80° to 94.56 ± 2.80°. The same observation was made by Wetzel et al. after electron beam sterilization of TPU [29]. Surface chemistry of the samples was analyzed by ATR-FTIR. Spectra, shown in Figure 5a, have been normalized relative to the 1077 cm^−1^ band. Few differences were observed between the control samples and the sterilized samples. In Figure 5b,c, changes can be observed for the 3326 cm^−1^ (N-H stretching vibration), 2940 cm^−1^ and 2850 cm^−1^ (C-H stretching vibration) bands. For the γ-sterilized sample: the 3326 cm^−1^ band presents a slight shoulder at 3300 cm^−1^, the 2940 cm^−1^ band begins to split and lastly for the 2852 cm^−1^ band, the peak slightly shifts towards 2850 cm^−1^. Mrad et al. evidenced that changes at 3326 cm^−1^, 2940 cm^−1^ and 2850 cm^−1^ are characteristic of ethylene bis stereamide (EBS), a European Pharmacopoeia approved lubricant, migrating at the surface of TPU samples by means of small-scale chain scissions [26].

### 3.3. Biological Properties of Sterilized 3DP Samples

#### 3.3.1. Cytotoxicity

The viability of cells is shown in Figure 6a. Compared with the control (100% cell viability), the polymer extracts of unsterilized samples and γ-sterilized samples exhibited cell viability of 151% ± 2% and 132% ± 3%, respectively. Therefore, as their relative cell viability far exceeding the non-cytotoxicity threshold of 70% defined in the ISO10993-5 standard, the irradiation of sterilization did not seem to negatively impact the non-cytotoxicity of the FDM processed material. The cell viability to sterilized materials was 12% lower than that non-sterilized samples, but without statistical significance (*p* > 0.05).

#### 3.3.2. Hemocompatibility

##### Hemolysis

The values of hemolysis rates (Figure 6b) calculated following the equation (ϕ) showed that, as expected, the positive control (i.e., 0.1% Na_2_CO_3_), yielded a 100% hemolysis while the negative control (i.e., PBS) did not yield any. Regarding test materials, with or without gamma sterilization, both groups exhibited no hemolytic activity (0%).

##### Blood Cells Adhesion

After exposure to human whole blood for 1 h, the surface of FDM fabricated samples (with or without γ-sterilization) were observed under SEM (Figure 6c). Although red blood cells were found adhered on the surface of both groups, the most abundant adhered blood cells onto the surface of material are platelets. Regarding the morphology of adhered platelets, on the γ-sterilized samples, they showed no visible pseudopodia; while on the non-sterilized FDM samples, flattened and spread platelets were found with abundant extended pseudopodia. This morphological transformation is considered as the hallmark of platelet activation, which consists first of adhesion of round platelets, then moderately shape-changed platelets or dendrites with pseudopodia, then followed by the flattening and the spreading of the platelets [30]. Therefore, the activation of adhered platelets was much more advanced on the surface of the non-sterilized FDM samples than that on the sterilized ones. It implies that the activation of platelets on TPU surface was diminished by gamma sterilization treatment. drawn.

## 4. Discussion

Our study aimed to evaluate the extent of degradation on TPU for the production of a medical device using a 3DP technique. The production route consisted of the processing of TPU by FDM from raw pellets to 3D printing and then the sterilization of the 3D constructs by γ irradiation at a dose of 40 kGy.

FDM was chosen for the availability of medical grade raw materials, the facility of use and adaptability for scaling. In the medical industry, it is required that the manufacturing process of a medical device is precisely monitored in order to achieve repeatedly the same quality and same properties for every item fabricated. We proved that FDM was a reproducible technique as we investigated repeatability through molecular weights of successively printed 3D constructs. Mn and Mw being constant after multiple successive printings enabled a validation of the printing parameters after a thorough optimization process on parameters such as the extrusion flow, the printing speed, the extrusion nozzle and bed temperatures (Appendix A). During the manufacturing, TPU underwent two successive processing steps with high shear stresses, i.e., extrusion and 3D printing, in order to obtain tubular 3D constructs. It has been evidenced that polymer degradation occurred at both steps as molecular weights decreased until ultimately reaching a cumulative loss of a third of Mw. Thermal analysis showed changes in glass transition temperatures, i.e., increase of Tg1 and decrease of Tg2. This pattern means that both successive processing steps could have introduced more phase mixing and a rearrangement between soft and hard domains of TPU. The dissolution of small or non-ideally arranged HS in soft domains could explain this raise in Tg1. Takahara A et al. explained that the degree of phase separation in the TPU between soft and hard domains increases when this temperature is closer to the Tg of the polyether chains [16]. Abraham et al. reported a Tg of −79 °C for an homopolymer of PTHF [27]. In the same fashion, for Tg2, HS domains in the raw pellets do not contain amorphous segments therefore cannot exhibit a glass transition temperature. With crystalline and amorphous phases reorganization occurring during the extrusion step, more phase mixing between HS and SS led to a detectable Tg2. Meijs et al. explained that this phenomenon takes place by means of small-scale chain scissions at the interface between hard and soft domains [31].

Sterility is required for a medical device to be implanted. Gamma sterilization was chosen as it is a common, highly effective, extensively documented method in the literature. The sterility assurance level required by the European Pharmacopoeia is obtained at a dose of 25 kGy [32]. The 3D printed samples were sterilized at a dose of 40 kGy. Sterilization entailed a macroscopic color change on FDM samples. Yellowing of irradiated samples is a reported consequence of oxidation of TPU as well as a decrease in hydrophobicity [24,29]. Several oxidation mechanisms for TPU following the type of energy input have been proposed including the conversion of the bis phenyl methylene groups into allenes groups with conjugated double bonds [33]. Free radicals generated by radiolysis with oxygen during irradiation could initiate these mechanisms [23] and the accumulation of conjugated double bonds from the formation of these chromophore structures is responsible for the color change [34]. Surface properties investigation via infrared spectroscopy indicated changes in surface chemistry after sterilization. The release of additives from the TPU surface into biological medium must be anticipated in the manufacturing of an implantable medical device therefore toxicity studies have to be carried out. Moreover, it has been evidenced in the literature by Murray et al. [28] that crosslinking happens by means of radical recombination. Tensile properties of FDM processed TPU after irradiation remain unchanged. One might think that with crosslinking happening, TPU would stiffen and loose flexibility. Murray et al. made the presumption that the effect of crosslinking in TPU are not recognizable with doses lower than 200 kGy. It has also been documented that commercial TPUs exhibit a resistance to irradiation up until a certain threshold of multiple hundreds of kGy [22,23,34]. By “only” undergoing a dose of 40 kGy, the elastomeric properties of TPU remained intact and compliant with our application.

In addition to physical and molecular behaviors, the biological properties of FDM processed then γ sterilized TPU must be adequate to the purpose that the medical device has been designed for. FDM samples before and after sterilization did not show hemolytic activity making them suitable for blood-contacting implants. Furthermore, not only FDM processed TPU exhibited non-cytotoxic behavior by greatly meeting the requirement for the extraction method, cell viability remained extremely high above 70% after irradiation. It is common knowledge that commercial TPUs are formulated with additives such as lubricants and antioxidants in order to achieve desired properties. We suggested earlier the possibility that EBS lubricant migrated on the surface of our FDM processed TPU samples after sterilization by means of small-scale chain scissions happening simultaneously with crosslinking. Bernard et al. studied the impact of this lubricant on the biological properties of cyclic olefin copolymers. For the grades containing it, they found that (i) cell viability, via MTT assay with HUVEC cells, was lowered from 175% to 150%; and (ii) less adhesion and platelet activation was observed [35]. Adhesion and activation of platelets are foundational steps involved in the occurrence of a thrombosis. Interestingly, we observed the same pattern in our case: cell viability was slightly lowered and platelet activation was seemingly reduced.

Biological properties of TPU were better after gamma sterilization considering the lesser platelet adhesion and lesser advanced activation on the surface of the sterilized 3DP TPU constructs. For further screening, in vitro bio-durability of FDM processed and γ-sterilized TPU must be evaluated, as a decrease of molecular weight observed during processing could lead to early loss of mechanical properties. In addition, the evolution of topology and surface porosity should be assessed over time to assess the impact on cell adhesion and proliferation [35].

## 5. Conclusions

For the design of a 3D printed and gamma sterilized blood-contacting implant, TPU appears to be a viable candidate, although FDM processing had a destructive effect on its molecular integrity occurring in successive processing steps, i.e., extrusion and 3D-printing. Sterilization using gamma rays seemingly induced crosslinking. Nevertheless, the mechanical properties of TPU were not impacted and biological properties were compliant with the application. Gamma-sterilized samples were found to be non-cytotoxic and non-thrombose inducing as platelet activation occurred to a lesser extent on the samples after gamma sterilization. Following FTIR analysis, we suspected the migration of a well-known lubricant used in TPU formulation, i.e., EBS, at the surface of the material. Its presence on the surface of the material will be investigated in a follow-up study. Ultimately, for this fabrication route to be approved from a risk-assessment based approach, the follow-up study should also involve the evaluation of bio-durability of TPU in vitro and analysis of leachable compounds.

## Figures and Tables

**Figure 1 pharmaceutics-15-00456-f001:**
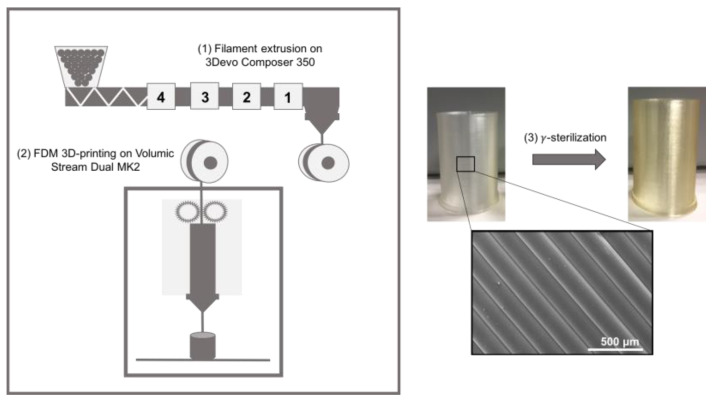
Production route of FDM processed TPU constructs.

**Figure 2 pharmaceutics-15-00456-f002:**
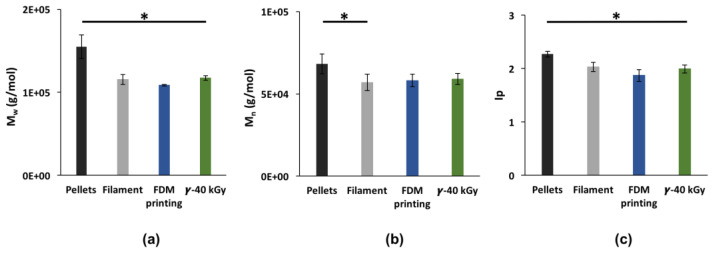
Impact of the manufacturing process on Mw (**a**); Mn (**b**); and Ip (**c**). Data are expressed as mean ± SD (n = 6), * *p* < 0.05 (between each processing step, Student *t*-test).

**Figure 3 pharmaceutics-15-00456-f003:**
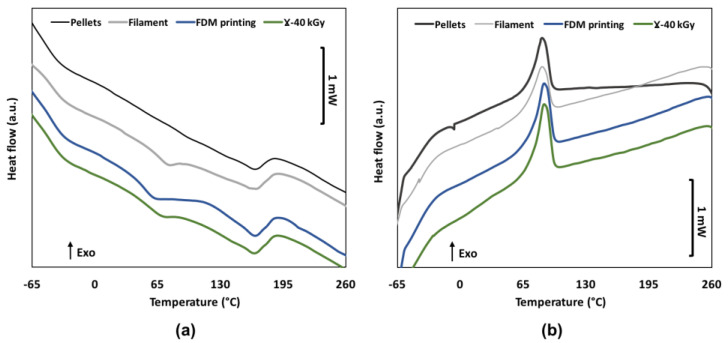
DSC traces from first heating run (**a**) and cooling run (**b**) for the raw pellets, extruded filament, FDM processed and sterilized samples.

**Figure 4 pharmaceutics-15-00456-f004:**
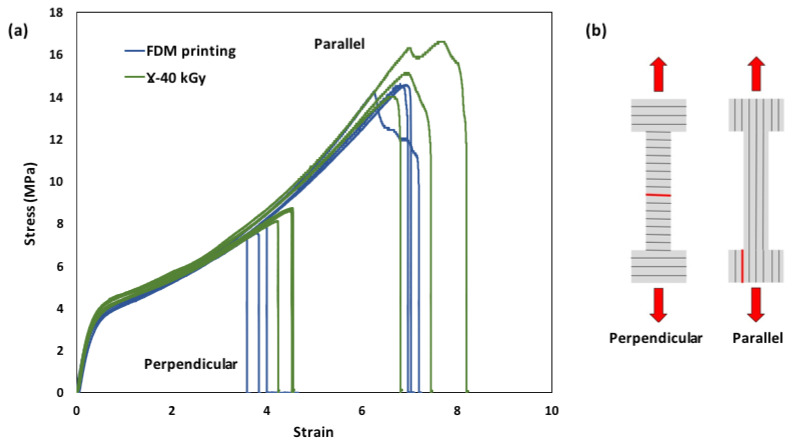
(**a**) Tensile curves for FDM processed and sterilized TPU samples; and (**b**) dumbbells cut in the perpendicular and parallel directions to the printing layers (red lines represent locations of breaking).

**Figure 5 pharmaceutics-15-00456-f005:**
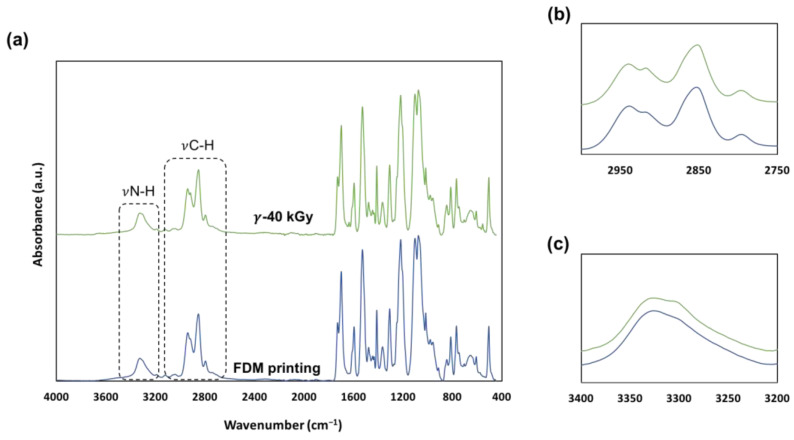
(**a**) ATR-FTIR spectra from 4000 cm^−1^ to 400 cm^−1^ of FDM processed and sterilized samples. (**b**) split at 2940 cm^−1^ and shift from 2852 to 2850 cm^−1^; and (**c**) shoulder appearance at 3300 cm^−1^.

**Figure 6 pharmaceutics-15-00456-f006:**
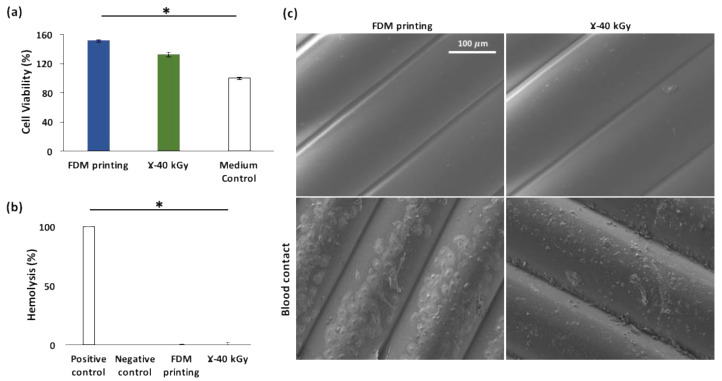
Impact of γ sterilization on the biological properties of FDM processed TPU constructs: cell viability (**a**); hemolysis rates (**b**); and SEM micrographs of FDM processed (without and without γ-sterilization) after blood contact (**c**) * *p* < 0.05 (compared to control, ANOVA one-way).

**Table 1 pharmaceutics-15-00456-t001:** Parameters for the FDM processing of TPU constructs.

Printing Parameters	Data
Flow (%)	130
Printing layer height (mm)	0.15
Nozzle Diameter (mm)	0.4
Nozzle temperature (°C)	220
Printing bed temperature (°C)	50
Printing speed (mm/min)	1000

**Table 2 pharmaceutics-15-00456-t002:** Collected DSC data (n = 3) from first cycle (mean ± SD).

Samples	Tg_1_ (°C)	Tg_2_ (°C)	T_m_ (°C)	ΔH_m_ (J/g)	T_c_ (°C)	ΔH_c_ (J/g)
Pellets	−50 ± 2	n.d.	164 ± 1	13.8 ± 0.5	84 ± 1	10.2 ± 0.4
Filament	−45 ± 1	64 ± 3	163 ± 1	11.2 ± 0.5	85 ± 1	9.0 ± 0.5
FDM printing	−48 ± 3	52 ± 2	163 ± 3	9.9 ± 3.4	88 ± 3	10.8 ± 0.7
Ɣ-40 kGy	−47 ± 4	61 ± 3	165 ± 1	7.3 ± 3.2	86.0 ± 0.4	11.0 ± 0.5

**Table 3 pharmaceutics-15-00456-t003:** Data (n = 3) from tensile curves: Young’s modulus (E), stress at break (σ_b_) and strain at break (ε_b_) (mean ± SD).

Samples	E (MPa)		σ_b_ (MPa)		ε_b_ (%)	
	Para	Perp	Para	Perp	Para	Perp
FDM printing	14 ± 1	14.0 ± 0.1	13 ± 2	8.0 ± 0.2	700 ± 20	380 ± 20
Ɣ-40 kGy	15.0 ± 0.5	14 ± 2	15 ± 1	8.0 ± 0.4	720 ± 70	440 ± 20

## Data Availability

Not applicable.

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
