# Peer review of "Evaluation of a Medical Grade Thermoplastic Polyurethane for the Manufacture of an Implantable Medical Device: The Impact of FDM 3D-Printing and Gamma Sterilization"

_pharmaceutics, 2023, doi:10.3390/pharmaceutics15020456_

Round 1
Reviewer 1 Report
1- Please mention the software used to design the 3D printed design.
2- Why did the authors use gamma sterilization specifically? Did the targetwas sterilization of surface modification through gamma irradiation? If the main purpose is sterilization so why not UV for example?
3- There is a typo mistake. It is nozzle not noozle.
4- Why did not the author provide SEM images for the 3D printed designs before any assessments?
5- Did the authors check the porosity? It is an important aspect for cells experiments.
Author Response
Authors thank reviewers for their attentive proof reading of our manuscript and for their judicious remarks that will strongly contribute to its amelioration. Our replies to their comments and questions are detailed point by point in their reports attached.

Reviewer 2 Report
The novelty of the article should be clearly added to the abstract.
Use quantitative results in the abstract. The abstract should be written more attractively. Most of it contains research methods.
On what basis are the parameters for granule extrusion (filament production) selected? Has a rheology or DMTA test been done? You can use this source for TPU printing (A New Strategy for Achieving Shape Memory Effects in 4D Printed Two-Layer Composite Structures; Development of Pure Poly Vinyl Chloride (PVC) with Excellent 3D Printability and Macro- and Micro-Structural Properties).
Why is 130 chosen from the floe value?
The nozzle diameter must be mentioned because one of the parameters that strongly affects the printing quality and buckling is the melt flow, which is affected by the nozzle diameter, printing speed, and layer thickness.
The number of repetitions of performed tests should be mentioned. Especially the mechanical properties.
Lines 185 to 189 are additional explanations and can be deleted.
The introduction could be more complete, especially in this field's FDM section and new materials. The following references are suggested to strengthen it. (4D Printing-Encapsulated Polycaprolactone–Thermoplastic Polyurethane with High Shape Memory Performances; 3D printing of PLA-TPU with different component ratios: Fracture toughness, mechanical properties, and morphology)
Printing one of the considered equipment and providing a printed image is better.
Some sections report entirely the results to which analysis should be added, for example, in sections 3.1 and 3.3.
Author Response

(The authors gave the same response as above.)
